# A mixed-methods study to explore the modifiable aspects of treatment burden in Parkinson's disease and develop recommendations for improvement

Qian Yue Tan[1,2,3]*, Kinda Ibrahim[1,2,4], Helen C. Roberts[1,2], Khaled Amar[5], Simon D.S. Fraser[2,4]

1 Academic Geriatric Medicine, Faculty of Medicine, University of Southampton, Southampton, United Kingdom, 2 National Institute for Health and Care Research Applied Research Collaboration Wessex, University of Southampton, Southampton, United Kingdom, 3 Portsmouth Hospitals University NHS Trust, Portsmouth, United Kingdom, 4 School of Primary Care, Population Sciences and Medical Education, Faculty of Medicine, University of Southampton, Southampton, United Kingdom, 5 University Hospitals Dorset NHS Foundation Trust, Bournemouth, United Kingdom

* q.y.tan@soton.ac.uk

## Abstract

### Background

People with Parkinson's (PwP) and their caregivers have to manage multiple daily healthcare tasks (treatment burden). This can be challenging and may lead to poor health outcomes.

### Objective

To assess the extent of treatment burden in Parkinson's disease(PD), identify key modifiable factors, and develop recommendations to improve treatment burden.

### Methods

A mixed-methods study was conducted consisting of: 1) a UK-wide cross-sectional survey for PwP and caregivers using the Multimorbidity Treatment Burden Questionnaire (MTBQ) to measure treatment burden levels and associated factors and 2) focus groups with key stakeholders to discuss survey findings and develop recommendations.

### Results

160 PwP (mean age = 68 years) and 30 caregivers (mean age = 69 years) completed the surveys. High treatment burden was reported by 21% (N = 34) of PwP and 50% (N = 15) of caregivers using the MTBQ. Amongst PwP, higher treatment burden was significantly associated with advancing PD severity, frailty, a higher number of

**Data availability statement:** All anonymised data are held in the University of Southampton Institutional Repository and can be accessed at https://eprints.soton.ac.uk/482384/.

**Funding:** This study was funded by the National Institute for Health and Care Research (NIHR) Applied Collaboration (ARC) Wessex. QYT received support from the NIHR ARC Wessex funded through the PhD fellowship award. QYT was supported by the University of Southampton NIHR Academic Clinical Fellow training programme. The funders had no role in study design, data collection and analysis, decision to publish, or preparation of the manuscript. The views expressed in this publication are those of the authors and not necessarily those of the National Health Service, the NIHR or the Department of Health and Social Care.

**Competing interests:** The authors have declared that no competing interests exist.

non-motor symptoms, and more frequent medication timings (>3 times/day). Caregivers reporting higher treatment burden were more likely to care for someone with memory issues, had lower mental well-being scores and higher caregiver burden. Three online focus groups involved 11 participants (3 PwP, 1 caregiver and 7 healthcare professionals) recruited from the South of England. Recommendations to reduce treatment burden that were discussed in the focus groups include improving communication. clear expectation setting, and better signposting from healthcare professionals, increasing education and awareness of PD complexity, flexibility of appointment structures, increasing access to healthcare professionals, and embracing the supportive role of technology.

## Conclusions

Treatment burden is common amongst PwP and caregivers and could be identified in clinical practice using the MTBQ. There is a need for change at individual provider and system levels to recognise and minimise treatment burden to improve health outcomes in PD.

## Introduction

Parkinson's disease (PD) is a common, progressive, incurable neurological condition causing multiple motor and non-motor symptoms [1]. The global prevalence of PD is increasing due to the rising ageing population [2]. Current management strategies for PD focus on achieving optimal symptomatic control through medical or surgical interventions, involving a multidisciplinary team of healthcare professionals [3,4]. People with Parkinson's (PwP) commonly have other long-term conditions such as osteoarthritis, hypertension or diabetes, thereby meeting the commonly accepted definition for multimorbidity [5]. Many also live with frailty, a 'distinctive state associated with ageing with increased vulnerability to stressors' that is reported to be prevalent in 23–57% of PwP [6]. As a result, PwP may have to take multiple medications, attend appointments with various healthcare professionals, learn about their health and make lifestyle changes to manage their health [7,8]. This workload of healthcare and the impact on a patient's well-being is termed 'treatment burden' [8].

Managing PD alongside multimorbidity and/or frailty can be burdensome for PwP and their caregivers. Previous studies conducted by our research team, including a qualitative systematic review and interviews, identified some of the main issues of treatment burden that PwP and their caregivers experienced when attempting to manage their health [9,10]. These include challenges attending appointments, difficulty accessing healthcare, issues obtaining and understanding information about PD, the workload of managing prescriptions and polypharmacy, and implementing recommended personal lifestyle adaptations. Treatment burden in PD is closely linked to 'capacity' – the ability to manage treatment burden [10,11]. The capacity of PwP and their caregivers may be influenced by various individualised factors, including the progressive physical and mental symptoms of PD, access and ability to use

a car or technology, availability of practical and emotional support from social networks, personal attributes, life circumstances and responsibilities [10]. Patients with long-term conditions who have high treatment burden and/or reduced capacity may experience negative outcomes such as poor quality of life, reduced ability to adhere to treatment regimens, and poor health outcomes [12].

To date, several validated tools have been developed to measure treatment burden. These include the Multimorbidity Treatment Burden Questionnaire (MTBQ) [13], Patient Experiences with Treatment and Self-Management (PETS) [14], and Treatment Burden Questionnaire (TBQ) [15]. However, none of these tools have been specifically used to quantify the treatment burden in PD, with limited research assessing the treatment burden experienced by caregivers [16]. Understanding the extent and key factors associated with high treatment burden for PwP and caregivers could inform changes required at individual provider and system levels to improve overall health outcomes in PD. Building on our previous research, this study aims to: 1) identify the extent and associations of treatment burden for PwP and caregivers; 2) identify key modifiable factors associated with treatment burden for PwP and caregivers, including the impact of multimorbidity and frailty; and 3) develop recommendations to improve the healthcare experiences of PwP and caregivers.

## Materials and methods

This study employed a sequential explanatory mixed-methods study design over two phases. Phase 1 consisted of a cross-sectional national survey of PwP and caregivers, and Phase 2 consisted of qualitative focus groups. Findings from our previous published qualitative studies that explored the treatment burden in PD informed the development of the national survey in Phase 1 [9,10]. The results from the national survey (Phase 1) were then integrated with findings from our previous qualitative studies to identify the key modifiable issues that impact treatment burden and capacity. This then informed Phase 2. The two phases are described further below.

Ethical approval for this study was obtained from the National Health Service Research Ethics Committee (21/WM/0058). The full study protocol is registered on Clinicaltrials.gov (NCT04769973). Written informed consent was obtained from participants. Licenses and permissions for the use of validated survey measures were obtained where required.

### Phase 1: Survey data collection and analysis

A cross-sectional **national survey** amongst PwP and caregivers living across all regions in the United Kingdom (UK) was conducted from September 2021 to January 2022. The aim of the survey was to determine the extent of treatment burden in PD and identify key associated factors. Adult participants (age > 18 years), with a diagnosis of PD, as well as caregivers for someone with PD, were recruited via two methods: 1) via Parkinson's UK Research Support Network and Take Part Hub, where a link to the participant information sheet and online surveys were advertised on the organisation's website, and 2) through two PD outpatient clinics in the South of England, where interested participants were given a survey pack containing the participant information sheet, survey booklet, and prepaid return envelopes. Two separate surveys, one for PwP and one for caregivers, were available in both paper and online formats (see S1 File) with matching questions wherever feasible. PwP were eligible irrespective of whether they had a caregiver. Similarly, caregivers could participate even if the person with PD they cared for was unable or did not want to. All data were self-reported. As this was an exploratory study, formal sample size calculation was not performed as the intention was not to establish causal relationships or predictive aspects of treatment burden in PD. All paper survey responses were manually double entered into the data spreadsheet by the first author (QYT), a research administrator and a medical student.

### Treatment burden measure

Treatment burden levels were measured using the MTBQ, which has been validated in older adults with multimorbidity in the UK [13]. For PwP, the 13-item MTBQ included difficulties related to taking multiple medications, self-monitoring, lifestyle

changes, obtaining health information, coordinating healthcare, impact on family and friends, financial burden, access to healthcare out of hours and access to community services. A 16-item MTBQ, adapted by the original research team but not yet validated, was used following direct communication and permission from the developers. The 16-item MTBQ included additional questions regarding difficulties arranging respite care, the financial impact of being a caregiver, and lifestyle adjustments to look after the person they cared for. Each item on the MTBQ was rated on a 5-point scale from '0' (not difficult/does not apply) to '4' (extremely difficult). Responses with >50% missing data were excluded from analysis [13].

## Other variables

Sociodemographic data collected included age, sex, ethnicity, marital status (married/civil partnership vs. single/divorced/dissolved civil partnership/widowed), living arrangements (living alone vs. living with spouse/partner/family) and employment status (employed vs. unemployed/retired). Variables related to treatment burden and capacity included the number of medications, frequency of medications, perceived difficulty getting PD-related information, and frequency of healthcare appointments for PD.

Health literacy was assessed using the single-item literacy score (SILS), scored on a 5-point Likert scale, where responses of 'some', 'often' and 'always' indicated limited health literacy [17]. Overall health and PD characteristics were assessed with length of PD diagnosis (years), PD severity (Hoehn and Yahr (H&Y) staging 1–5, with higher stage indicative of worse PD severity) [18], presence of non-motor symptoms (Non-Motor Symptoms Questionnaire (NMSQuest)) [19], self-reported number of other long-term conditions, frailty measure (PRISMA-7) [20], and health-related quality of life (Short-Form 12-Item Survey version 2)(SF12v2) [21] producing physical component summary (PCS) and mental component summary (MCS) scores. Caregiver burden was measured using the 12-item Zarit Burden Interview (ZBI-12) [22] with higher total scores indicative of higher caregiver burden levels. Caregiver burden is related to the emotional, social, financial, physical and spiritual impact and well-being of being a caregiver, and is a separate concept that may impact caregiver treatment burden [10,23].

## Quantitative statistical analysis

Statistical analysis was conducted using the IBM Statistical Product and Service Solutions (SPSS) software. Surveys completed by PwP and caregivers were analysed independently. No comparisons were made between online and paper survey responses due to the small number of paper surveys (N = 31). Global MTBQ scores were calculated as per Duncan et al and categorised into 'no burden' (score = 0), 'low burden (score <10)', 'medium burden' (score 10–21) and 'high burden' (score ≥22) [13]. Participants with no and low treatment burdens were combined into one group, and those with medium and high treatment burdens were combined into another group for analysis. This dichotomisation was felt to be clinically appropriate given the progressive nature of PD, where early recognition of PwP and caregivers with medium and high treatment burden could allow timely interventions to prevent increasing treatment burden or reduce treatment burden. Other studies using the MTBQ have used similar statistical approaches to dichotomisation using high vs no/low/medium treatment burden levels [24,25].

Descriptive statistics were used and presented as median (interquartile range (IQR)), mean (standard deviation (SD)) and number (%) as appropriate. For PwP, univariable and multivariable binary logistic regression analysis were conducted using the dichotomised MTBQ scores (no/low vs medium/high burden). Variables included in the multivariable logistic regression model were decided a priori based on known associations with treatment burden from previous studies (age, number of medications, number of long-term conditions), those hypothesised to be clinically relevant after discussion within the research team, and variables that achieved p < 0.25 at univariable analysis. Univariable pre-screening is an approach that can be used to determine inclusion of variables into the regression model using a less stringent p-value (p < 0.25 in this study) [26]. This is an established approach to ensure that potentially important confounders are not excluded prematurely, thereby preserving the validity of the final model [26,27].

Due to the smaller sample size, regression analysis was not possible for caregivers. Instead, participant characteristics for caregivers in the 'no/low' burden vs. 'medium/high' burden groups were compared using the independent t-tests, Mann-Whitney U tests, Pearson Chi-square tests, Fischer's exact tests, or likelihood ratio tests according to the distributional properties of each variable.

### Phase 2: Focus group data collection and analysis

Findings from the national surveys were integrated with results from our previous qualitative studies to identify the key issues of treatment burden and capacity in PD [9,10]. These relate to attending appointments and interactions with healthcare professionals, satisfactory information provision, managing prescriptions and medications, and personal life adaptations. These key issues informed the development of the focus group guide (see S2 File), with the use of open-ended questions to facilitate discussions of potential strategies to reduce the treatment burden for PwP and caregivers. A purposive sample of key stakeholders in the South of England were invited to participate in online focus groups. These included PwP, caregivers and healthcare professionals involved in the care of PD such as PD specialists, PD nurse specialists, general practitioners, psychiatrists, pharmacists, physiotherapists, and occupational therapists. Interested PwP and caregivers from outpatient PD clinics were approached after their clinic appointment, following consent from their clinician, and provided a participant information sheet. Healthcare professionals were recruited from local services, the Parkinson's UK Wessex regional network and the professional networks of the research team.

All consenting participants received a one-page summary of the identified key issues of treatment burden and capacity in PD prior to the focus group session. Each focus group began with an overview of the study, a reminder of the discussion aims, and agreed ground rules such as confidentiality, listening and respecting others. Three focus groups were conducted on Microsoft Teams between May and July 2022 and recorded with written consent. The discussions were moderated by QYT, a female geriatrician with a specialist interest in PD. Automatic transcription software within Microsoft Teams was used, and transcripts were anonymised and manually corrected by QYT before thematic analysis [28]. QYT took written notes, listened to the recordings, edited transcriptions, and read the transcriptions multiple times to enable immersion in the data. Inductive coding was conducted based on the issues of treatment burden discussed. Mind maps were then used to determine the connections and interlinks between the codes which led to the overall themes of recommendations for improvement. This was then reviewed and refined with the wider research team.

### Patient and public involvement

The study involved a patient and public involvement group of a caregiver of someone with PD and a caregiver of someone with dementia and other long-term conditions in the study design, review of protocol and all patient-facing documents. The national surveys and focus group guide were piloted with PwP and caregivers prior to finalisation. This feedback led to revisions to ensure the survey captured all aspects of treatment burden and capacity experienced in PD, reduced survey burden and improved ease of understanding.

### Results

### Phase 1: Survey findings

The full characteristics of survey participants are presented in Table 1. A total of 162 PwP (143 online, 19 paper) and 30 caregiver (18 online, 12 paper) surveys were completed. Two PwP survey responses were excluded due to >50% missing data on the MTBQ, leaving 160 valid responses. The mean ages of PwP and caregivers were 67.6±9.2 and 68.7±8.9 years, respectively. Among PwP, 52% (N=83) of respondents were females, whilst 73% (N=22) of caregivers were females. Most participants were married, cohabiting with a spouse/partner/family member, living in their own home, retired, and had at least GCSE-level education. PwP reported a median disease duration of 5 (IQR 3–8) years, whilst

**Table 1. Characteristics of survey participants.**

| Variables | | PwP (N = 160) | Caregivers (N = 30) |
|---|---|---|---|
| Mean age, years (SD) | | 67.6 (8.2) | 68.7 (8.9) |
| Gender, N (%) | Male | 76 (48%) | 8 (27%) |
| | Female | 84 (52%) | 22 (73%) |
| Ethnicity, N (%) | *Missing data* | 1 (1%) | – |
| | White | 158 (98%) | 30 (100%) |
| | Non-white | 1 (1%) | – |
| Marital status, N (%) | *Missing data* | 1 (1%) | – |
| | Married/ civil partnership | 126 (79%) | 28 (94%) |
| | Single | 11 (7%) | 1 (3%) |
| | Divorced/ dissolved civil partnership | 15 (9%) | 1 (3%) |
| | Widowed | 7 (4%) | – |
| Living situation, N (%) | *Missing data* | 1 (1%) | – |
| | Alone | 21 (13%) | 1 (3%) |
| | With spouse/partner/family | 138 (86%) | 29 (97%) |
| Living property, N (%) | Own | 140 (88%) | 29 (97%) |
| | Rented/Other | 20 (12%) | 1 (3%) |
| Employment status, N (%) | Retired | 126 (79%) | 25 (83%) |
| | Employed | 25 (16%) | 3 (10%) |
| | Unemployed | 9 (3%) | 2 (7%) |
| Highest education level, N (%) | *Missing data* | 1 (1%) | 1 (3%) |
| | Degree level or above | 91 (57%) | 8 (27%) |
| | A level or equivalent | 35 (22%) | 10 (33%) |
| | GCSE level or equivalent | 23 (14%) | 8 (27%) |
| | No qualification | 10 (6%) | 3 (10%) |
| Median length of PD diagnosis, years (IQR)* *Missing data =* | | 5 (3-8) 6 | 10 (6-15) 1 |
| PD severity (H&Y stage)* | *Missing data* | – | 1 (3%) |
| | Mean (SD) | 2.0 (1.0) | 3.1 (1.4) |
| | Stage 1 | 76 (47%) | 7 (23%) |
| | Stage 2 | 20 (13%) | 1 (3%) |
| | Stage 3 | 55 (34%) | 9 (30%) |
| | Stage 4 | 8 (5%) | 7 (23%) |
| | Stage 5 | 1 (1%) | 5 (17%) |
| Median PD NMSQuest score, (IQR) | | 9 (6-13) | |
| Caregiver reported symptoms in the person with PD in the last 12 months, N (%) | Mood | | 22 (73%) |
| | Memory | | 22 (73%) |
| | Hallucinations | | 15 (50%) |
| PwP reported number of other long-term conditions, N (%)* | *Missing data* | 18 (11%) | 8 (27%) |
| | 0-1 | 69 (43%) | 3 (10%) |
| | ≥2 | 73 (46%) | 19 (46%) |
| Frailty, N (%) | Yes | 74 (46%) | 4 (13%) |
| | No | 86 (54%) | 26 (87%) |
| Mean Physical Component Summary (PCS) score (SD) | | 44.2 (10.3) | 49.6 (11.4) |
| Mean SF12v2 Mental Component Summary (MCS) score (SD) | | 47.2 (9.7) | 44.1 (10.5) |
| Median ZBI-12 score (IQR) | | 18.5 (8.8-27.5) | |

*(Continued)*

Table 1. (Continued)

| Variables | | PwP (N = 160) | Caregivers (N = 30) |
|---|---|---|---|
| MTBQ scores | 0 (No Burden) | 25 (16%) | 6 (20%) |
| | <10 (Low Burden) | 48 (30%) | 6 (20%) |
| | 10-21 (Medium Burden) | 53 (33%) | 3 (10%) |
| | ≥ 22 (High Burden) | 34 (21%) | 15 (50%) |
| Median number of medications taken by PwP (IQR)* Missing data = | | 4 (2-7) 3 | 6 (4-9) – |
| Number of medications taken by PwP, N (%) | Missing data | 3 (2%) | – |
| | 0-1 medications | 13 (8%) | 3 (10%) |
| | 2 medications | 30 (19%) | 2 (7%) |
| | 3 medications | 15 (9%) | 2 (7%) |
| | 4 medications | 26 (16%) | 2 (7%) |
| | ≥ 5 medications | 73 (46%) | 21 (69%) |
| Median frequency of medication times a day (IQR)* Missing data = | | 4 (3-5) 4 | |
| PwP requiring help with medications, N (%)* | Missing data | 3 (2%) | – |
| | Yes | 22 (14%) | 20 (67%) |
| | No | 135 (84%) | 10 (33%) |
| Health literacy, N (%) | Missing data | 1 (1%) | 1 (3%) |
| | Limited | 17 (11%) | 3 (10%) |
| | Not Limited | 142 (88%) | 26 (87%) |
| Level of difficulty getting information about PD, N (%) | Missing data | 1 (1%) | 1 (3%) |
| | Very Easy | 29 (18%) | 5 (17%) |
| | Easy | 66 (41%) | 12 (40%) |
| | Neither easy nor difficult | 50 (31%) | 10 (33%) |
| | Difficult | 12 (8%) | 1 (3%) |
| | Very Difficult | 2 (1%) | 1 (3%) |
| Median total number of contacts with healthcare services for PD in the last 12 months, (IQR)* | | 4 (2-8) | 6 (2-6) |

H&Y; Hoehn and Yahr, IQR; interquartile range, NMSQuestion; Non-Motor Symptoms Questionnaire, SD; standard deviation, SF12v2; Medical Outcomes Study Short Form version 2, *Caregiver reported regarding the person with PD they care for

caregivers looked after someone with a median disease duration of 10 (IQR 6–15) years. Using the H&Y staging severity, most PwP (94%) had early to mid stages of PD (stages 1–3), compared to most caregivers (70%) who reported caring for someone with mid to late stages of PD (stages 3–5). PwP reported a median number of 9 (IQR 6–13) non-motor symptoms on the NMSQuest. Most caregivers reported that the person with PD they cared for had problems with mood (73%), memory (73%) and hallucinations (50%) in the last 12 months. Nearly half of PwP had frailty (46%) and multimorbidity (46%).

High treatment burden was reported by over one-fifth of PwP (21%) and half of caregivers (50%). For PwP, aspects of treatment burden on the MTBQ reported as most difficult (responses 'a little to extremely difficult') were making recommended lifestyle changes (51%), remembering how and when to take medication (49%), obtaining clear and up to date information (49%) and arranging appointments with health professionals (48%). For caregivers, the most 'difficult' aspects on the MTBQ were adjusting their own lifestyle to look after the person they cared for (69%), making recommended changes to their lifestyle (59%), seeing lots of different health professionals (53%), arranging appointments with health professionals (50%) and getting help from community services (50%). Full data can be seen in S3 File.

## Associations with medium/high treatment burden

Univariate analysis found that PwP with higher H&Y stages (vs H&Y Stage 1) (H&Y stage 2 (odds ratio (OR) 1.45; 95% confidence interval (CI) 0.54–3.90); H&Y stage 3 (OR 3.54; 95% CI 1.69–7.42); H&Y stages 4 and 5 (OR 5.08; 95% CI 0.99–25.11), higher NMS scores (OR 1.13; 95% CI 1.06–1.21), frailty (OR 3.08; 95% CI 1.61–5.92), and those who took medications more than three times a day (OR 3.42; 95% CI 1.68–6.95) had significantly ($p < 0.05$) higher odds ratios of medium/high treatment burden. Multivariate analysis adjusting for age, gender, living property and employment for PwP found that PD severity (vs H&Y stage 1) (H&Y stage 2 (OR 1.56; 95% CI 0.56–14.36); H&Y stage 3 (OR 3.60; 95% CI 1.63–7.499); H&Y stages 4 and 5 (OR 3.93; 95% CI 0.72–21.48)), PD NMSQuest score (OR 1.12; 95% CI 1.04–1.21), frailty (OR 3.12; 95% CI 1.46–6.67), and frequency of medications (>3 times a day) (OR 3.01; 95% CI 1.44–6.30) had significant, independent associations ($p < 0.05$) with medium/high treatment burden. PwP with multimorbidity had higher odds of medium/high treatment burden (OR 1.45; 95% CI 0.75–2.93) although this was not statistically significant. These results are shown in Table 2.

**Table 2. Associations of treatment burden in people with Parkinson's.**

| Variables | | Univariate analysis | | | Multivariate analysis model* | | |
|---|---|---|---|---|---|---|---|
| | | OR | 95% CI | P value | aOR | 95% CI | P value |
| **Age** *(continuous variable)* | | 0.99 | 0.95–1.03 | 0.60 | | | |
| **Gender** *(vs female)* | | 1.61 | 0.86–3.01 | 0.14 | | | |
| **Marital status** *(vs married/civil partnership)* | Single or divorced/dissolved civil partnership or widowed | 1.16 | 0.53–2.51 | 0.71 | | | |
| **Living situation** *(vs alone)* | With spouse/ partner/ family or friends | 0.71 | 0.28–1.83 | 0.48 | | | |
| **Living property** *(vs own home)* | Rented or in family/ friends' property | 2.83 | 0.98–8.22 | 0.06 | | | |
| **Employment** *(vs unemployed or retired)* | Employed | 0.50 | 0.21–1.20 | 0.12 | | | |
| **Length of PD diagnosis** *(years)* | | 1.05 | 0.98–1.13 | 0.19 | 1.05 | 0.97–1.14 | 0.22 |
| **PD severity (H&Y stage)** *(vs stage 1)* | Stage 2 | 1.45 | 0.54–3.90 | 0.004 | 1.56 | 0.56–4.36 | 0.01 |
| | Stage 3 | 3.54 | 1.69–7.42 | | 3.60 | 1.63–7.94 | |
| | Stages 4 and 5 | 5.08 | 0.99–26.11 | | 3.93 | 0.72–21.48 | |
| **PD NMSQuest score** | | 1.13 | 1.06–1.21 | <0.001 | 1.12 | 1.04–1.21 | 0.002 |
| Other long-term conditions (vs 0–1) | ≥2 long-term conditions | 1.55 | 0.83–2.91 | 0.17 | 1.48 | 0.75–2.93 | 0.26 |
| Frailty (vs not frail) | Frail | 3.08 | 1.61–5.92 | <0.001 | 3.12 | 1.46–6.67 | 0.003 |
| Quality of life (SF12v2) | Physical component score | 1.00 | 0.97–1.03 | 0.83 | | | |
| | Mental component score | 0.99 | 0.95–1.02 | 0.38 | | | |
| Number of medications (vs 0–1) | 2 | 2.40 | 0.63–9.12 | 0.39 | 1.50 | 0.36–6.26 | 0.40 |
| | 3 | 1.07 | 0.23–4.89 | | 0.63 | 0.12–3.31 | |
| | 4 | 3.02 | 0.76–12.00 | | 2.49 | 0.57–10.82 | |
| | ≥5 | 1.94 | 0.60–6.50 | | 1.43 | 0.39–5.28 | |
| Frequency of medications (vs 0–3 times a day) | >3 times a day | 3.42 | 1.68–6.95 | <0.001 | 3.01 | 1.44–6.30 | 0.003 |
| Health literacy (vs not limited) | Limited | 2.99 | 0.93–9.60 | 0.07 | 3.26 | 0.98–10.83 | 0.054 |
| Total healthcare service use for PD in the last 12 months (vs 0–2) | ≥3 times | 1.14 | 0.58–2.25 | 0.70 | | | |

*aOR; Adjusted Odds Ratio, CI; Confidence Interval, H&Y; Hoehn and Yahr, NMSQuest; Non-Motor Symptom Questionnaire*

*Model adjusted for age, gender, living property and employment.

Descriptive analyses comparing those with no/low and medium/high treatment burden for caregivers are shown in Table 3. Caregivers who reported medium/high treatment burden were predominantly female, younger age, more likely to care for someone with PD H&Y stages 4–5, more likely to report memory issues in the person with PD they cared for, had lower mean mental component scores and higher caregiver burden levels compared to caregivers with no/low treatment burden.

**Table 3. Comparison between caregivers of people with Parkinson's with no/low and medium/high burden.**

| Variables | | No/Low Burden (N=12) | Medium/High Burden (N=18) | P value‡ |
|---|---|---|---|---|
| Mean age (SD), years | | 71.4 (6.1) | 66.5 (10.1) | 0.07* |
| Gender, N (%) | Male | 6 (50%) | 2 (11%) | 0.03 |
| | Female | 6 (50%) | 16 (89%) | |
| Marital status, N (%) | Single (never married or in a civil partnership) | 0 | 1 (6%) | 0.34‡‡ |
| | Married or in a civil partnership | 12 (100%) | 16 (89%) | |
| | Widowed | 0 | 1 (6%) | |
| Living situation, N (%) | Alone | 0 | 1 (6%) | 1.00 |
| | With spouse/partner or family member | 12 (100%) | 17 (94%) | |
| Living property, N (%) | Own property | 12 (100%) | 17 (94%) | 1.00‡‡ |
| | Relative's Home | 0 | 1 (6%) | |
| Employment status, N (%) | Employed | 0 | 3 (17%) | 0.06‡‡ |
| | Unemployed | 0 | 2 (11%) | |
| | Retired | 12 (100%) | 13 (72%) | |
| Highest education level, N (%) | Degree level or above | 1 (9%) | 7 (39%) | 0.06‡‡ |
| | A level or equivalent | 3 (27%) | 7 (39%) | |
| | GCSE level or equivalent | 6 (55%) | 2 (11%) | |
| | No qualification | 1 (9%) | 2 (11%) | |
| Median length of PD diagnosis, years (IQR) | | 9 (3-15) | 10 (7.75-14) | 0.55† |
| PD severity (H&Y stage), N (%) | Mean (SD) | 2.6 (1.6) | 3.3 (1.3) | 0.31* |
| | Stage 1 | 4 (36%) | 3 (17%) | |
| | Stage 2 | 1 (9%) | 0 | |
| | Stage 3 | 3 (27%) | 6 (33%) | |
| | Stage 4 | 1 (9%) | 6 (33%) | |
| | Stage 5 | 2 (18%) | 3 (17%) | |
| Caregiver reported presence of symptoms in PwP, N (%) | Mood | 8 (67%) | 14 (78%) | 0.68 |
| | Memory | 6 (50%) | 16 (89%) | 0.034 |
| | Hallucinations | 5 (42%) | 10 (59%) | 0.36 |
| Median PwP number of long-term conditions other than PD, (IQR) | | 2 (2-4.75) | 2 (2-3.5) | 0.87† |
| Quality of life (SF12v2) | Mean PCS (SD) | 53.1 (6.6) | 47.4 (13.3) | 0.07* |
| | Mean MCS (SD) | 50.2 (10.0) | 39.9 (8.8) | 0.004* |
| Median ZBI-12 score (IQR) | | 10 (3.25-13.75) | 23 (17.5-29) | <0.001† |
| Median number of medications for PwP (IQR) | | 5.5 (3-10) | 6 (4.75-8.25) | 0.76† |
| Median total number of healthcare service use for PD in the last 12 months (IQR) | | 4.5 (2-7.75) | 6.5 (2-11.5) | 0.39† |
| Health literacy, N (%) | Not Limited | 11 (100%) | 15 (83%) | 0.27 |
| | Limited | 0 | 3 (17%) | |

‡*Fisher's exact test unless otherwise stated; ‡‡Likelihood ratio; *Independent t-test; †Mann-Whitney U test; H&Y; Hoehn and Yahr, IQR; Interquartile Range; NMS, Non-Motor Symptoms; SD, Standard Deviation; SF12v2, Medical Outcomes Study Short Form version 2; ZBI, Zarit Burden Interview*

## Phase 2: Focus group findings

Three online focus groups were conducted with a total of 11 participants, comprising three PwP, one caregiver and seven healthcare professionals (see Table 4). The recommendations of ways to reduce treatment burden and/or increase patient and caregiver capacity were categorised into four main themes which are summarised in Table 5, along with descriptions of how each recommendation may improve specific aspects of treatment burden and capacity in PD.

**Theme 1: Visibility of Parkinson's.**  Participants with PD discussed the potential benefits of increasing visibility of the condition through positive labelling ("I have Parkinson's). This was discussed as a way to prioritise access to healthcare professionals for PwP and caregivers, increase awareness and education about the complexity of PD symptoms and recognise that not all symptoms are attributed to PD. The increased visibility of PD could enable appropriate advice or signposting to services and ensure delivery of proactive and holistic patient-centred care. In hospital settings, this could ensure PD medications are correctly prescribed and given on time.

**Theme 2: Improving availability and organisation of healthcare services.**  Participants described the current rigid structure of healthcare appointments and how increasing flexibility of appointment structures (length of appointments, time between appointments, mode of appointment (face-to-face or virtual) based on patient complexity and needs could improve this. Patient-initiated follow-up appointments or group appointments for people at early stages of PD were also suggested. The potential role of a single-point-of-access service led by an appropriately trained clinical administrator or wider member of the multidisciplinary team that could signpost PwP and caregivers to the most appropriate resource or services (PD specialist, PD nurse specialist, General Practitioner, or pharmacist), early referral and access to physiotherapy services, and virtual ward multidisciplinary approach for PwP and caregivers with complex needs within the community were suggested to enable proactive care. Improving communication between healthcare services, shared online medical records and regular forums for healthcare professionals within the regional health services could improve care coordination and lead to improved experiences of navigating health services for PwP and caregivers. The potential use of UK National Health Service (NHS) primary care prescription forms (FP10) or access to electronic prescriptions by PD specialists could minimise errors with adjusting PD medications. This was recommended with caution due to potential drug interactions if full access to the patient's medication history or shared medical records is unavailable. Furthermore, the role of community pharmacists or pharmacy technicians to help support PwP and caregivers enact PD medication changes was discussed.

**Table 4.  Focus group participants.**

| Focus Group Number | Participants | ID |
|---|---|---|
| FG1 | Person with PD | P01 |
| FG1 | Caregiver for person with PD | P02 |
| FG1 | PD specialist doctor | P03 |
| FG1 | PD specialist doctor | P04 |
| FG2 | Person with PD | P05 |
| FG2 | Person with PD | P06 |
| FG2 | Community clinical pharmacist | P07 |
| FG2 | Community clinical pharmacist | P08 |
| FG3 | PD specialist doctor | P09 |
| FG3 | Consultant old age psychiatrist | P10 |
| FG3 | Community physiotherapist | P11 |

*FG, Focus group; PD, Parkinson's disease*

**Table 5. Focus group themes and suggested recommendations.**

| Theme | Subthemes | Supportive quotes | Suggested recommendations that could lead to improvement of treatment burden |
|---|---|---|---|
| **Theme 1: Visibility of Parkinson's** | "I have Parkinson's" | *"I think it would be nice to have that badge really so that you get a priority… I would be happy to have that on my shirt." P05, PwP* | • Visibility of PD diagnosis as a key to ensure prioritised access to healthcare professionals<br>• Timely and accurate access to PD medications in hospital |
| | Improving education and awareness about Parkinson's | *"I always feel that people with Parkinson's get a really rough deal because as soon as they're diagnosed with Parkinson's, any symptom, they go to anybody with is labelled as "It's your Parkinson's."" P04, PD specialist* | • Recognition and awareness of PD symptoms and complexity to address issues with appropriate access to specialists<br>• Improvement of proactive care with increased healthcare professionals' awareness of PD complexity |
| **Theme 2: Improving availability and organisation of healthcare services** | Improving healthcare service capacity | *"The other thing that you can do is go towards a patient-initiated follow-up… So if patients don't want to have frequent follow-ups, they can say, "I don't want that appointment in nine months. I'd rather it be a year."" P03, PD specialist* | • Increased flexibility of appointment structures and potential use of patient-initiated follow-up or group appointments<br>• Use of single-point access to help signpost and improve access<br>• Use of virtual wards with input from a multidisciplinary team |
| | Improving care coordination between healthcare services | *"I think it would be nice if we had some sort of regular forum, if only just to familiarise ourselves with who we know, who we are and what we do. And get started to get some informal general advice, if we can progress that to specific case discussions about challenging patients that will be fantastic." P10, Psychiatrist*<br><br>*"In an ideal world, what I would then like to happen is obviously when that (PD) clinic letter is read in a GP practice, somebody will then contact you again to reiterate the same information. And that is what we're trying to work towards." P07, Pharmacist* | • Shared medical records, improving speed of communication and regular multidisciplinary forums for healthcare professionals involved in PD could improve care coordination<br>• Use of FP10 prescription by PD specialist or online prescription changes to reduce prescription delays or errors<br>• Access to enhanced support from primary care pharmacists to support medication changes in PD<br>• Early referrals and access to physiotherapy from early-stage PD to iterate the importance of physical activity |
| **Theme 3: Improving interactions with healthcare professionals and information provision** | Clear communication and setting expectations | *"And this whole normalizing it. Trying to persuade my parents that some of the things my father is struggling with are A: due to the Parkinson's, and B: completely normal for somebody with Parkinson's is extremely helpful because it's so difficult to get them to accommodate." P02, Caregiver*<br><br>*"I write to the GP, copy to the patient and then copy to (PD nurse specialist) plus to any other health professionals who've been directly involved. And I try and explain all my terms in brackets." P09, PD specialist* | • The normalisation of PD symptoms and expectation setting can enable shared-decision making and improve interactions between PwP, caregivers and healthcare professionals<br>• Clear explanation and communication during PD clinic appointments, including the use of clinic letters to PwP in lay terms to communicate outcomes and advice from appointments |
| | Opportunity to signpost towards information and services | *"So yeah, I think everyone using Parkinson's UK as a kind of national resource. It's good to have kind of central point so that everybody is using the same information." P07, Pharmacist* | • Signposting based on personal preferences, symptoms experienced, information to help medication management, local support groups or voluntary services to support living with PD<br>• Recognition of health literacy and appropriate signposting based on this |
| **Theme 4: Embracing the role of technology** | Video appointments or smartphone applications for review of Parkinson's | *"I mean just using Microsoft Teams, the program we're talking on now, or something like it. So you don't have to see everyone face-to-face. But you know, I think we could make better use of technology, to you know shorten the problems between healthcare professionals and patients." P06, PwP*<br><br>*"They need to look at the videos as well to help have a better understanding cause on paper, it's really difficult to explain the movement." P11, Physiotherapist* | • Improve access to healthcare professionals through the use of telephone or video appointments<br>• Support medication-taking by using reminders on devices such as smartphones or smartwatches<br>• Use of videos to demonstrate recommended exercises |

*P, Focus group participant ID; PD, Parkinson's disease; PwP, People with Parkinson's

**Theme 3: Improving interactions with healthcare professionals and information provision.** Participants discussed the importance of clear communication and realistic expectation settings by healthcare professionals, using clinical encounters as an opportunity to signpost PwP and caregivers towards appropriate information and services. Acknowledging and normalising the symptoms of PD were discussed as reassuring and could help PwP and caregivers manage their health better. Improvement in written communication, with clinic letters either addressed to PwP and caregivers or copying them into the letter with lay summaries explaining medical terms could help reiterate information discussed during appointments. Providing individualised information and signposting to services based on personal preferences, condition-specific leaflets on PD symptoms such as constipation and anxiety, or leaflets regarding living aids to support medication management was agreed amongst participants. The provision of a local resource of information that can be used to signpost PwP and caregivers to the services and activities available within their specific locality was also discussed.

**Theme 4: Embracing the role of technology.** Digital solutions such as telephone or video clinical consultations based on patient preference and accessibility the use of smartphone applications to monitor symptoms and record PD medications that could be shared with PD specialists for review and the use of smartphones or smartwatch reminders to support medication adherence were discussed as potential ways to reduce treatment burden. Video-based exercises targeted at the appropriate level were seen as appropriate alternative to paper descriptions and could help support recommended physical activities in PD.

## Discussion

This is the first study to use the MTBQ to demonstrate that treatment burden is common in PD, with over one-fifth of PwP and half of caregivers experiencing high treatment burden levels. Amongst PwP, higher treatment burden levels were associated with those living with frailty, more advanced stages of PD, a higher number of non-motor symptoms, and higher medication frequency. Making recommended lifestyle changes was the most difficult aspect of treatment burden experienced by PwP and caregivers, with similar findings reported in UK studies involving patients with multimorbidity [13,24].

Nearly half of PwP in this study reported having multimorbidity, with similar proportions living with frailty. Although multimorbidity was not associated with higher treatment burden levels in PwP, the use of self-reported data for other long-term conditions in this survey may have large variations with general practice medical records and could explain this finding [29]. The lack of association between multimorbidity and treatment burden was similarly reported in a study using the PETS treatment burden measure amongst patients undergoing dialysis treatments [30]. Conversely, other studies in older adults with multimorbidity have reported positive associations between the number of long-term conditions and treatment burden levels measured using the MTBQ [13,24]. The impact of multimorbidity on treatment burden in PD requires further evaluation. To our knowledge, the association between frailty and treatment burden has not been explored in other studies and is therefore a novel finding. However, this independent association in PwP should be interpreted with caution due to the potential overlap between frailty and the underlying neurodegenerative process in PD, which can affect physical function through fatigue and reduced walking speed [6]. Although no frailty measures have been validated in PD, active screening and early recognition are crucial, as there may be potential interventions that can improve health outcomes for PwP living with frailty [31]. Furthermore, the British Geriatrics Society has recommended the minimisation of treatment burden, particularly for older people living with advanced frailty, emphasising the importance of open conversations around the benefits and burdens of continuing active health interventions [32].

Caregiver treatment burden remains an under-researched concept in the literature, with no previous studies conducted specifically in PD despite the fundamental role they have in supporting the person with PD [16]. Our study reports

a substantial proportion of caregivers with high treatment burden levels. Caregivers with medium/high treatment burden were more likely to care for someone with more advanced stages of PD and for those who may have cognitive concerns. They also had higher odds of reporting lower mental health well-being scores and higher caregiver burden. The most difficult aspect of treatment burden for caregivers was having to adjust their own lifestyle to look after the person with PD, followed by supporting the person with PD to make recommended lifestyle changes. Studies amongst caregivers of older adults who have multimorbidity, diabetes and dementia have reported similar challenges supporting dietary changes and managing mealtimes [33,34]. Our survey findings perhaps reflect the change in personal role and limitations of activity for caregivers, which may be worsened when memory issues in the person they care for occur as PD progresses. Recognition that caregivers of PwP may experience the same, if not higher levels of treatment burden when supporting someone with PD is important, as caregiver treatment burden may also impact health outcomes for PwP.

The focus groups involved multiple key stakeholders and generated potential recommendations (see Table 5) at both individual and system levels that could be mapped across the main aspects of treatment burden and capacity for PwP and caregivers [9,10]. Flexibility of appointment structures, use of telemedicine and patient-initiated follow-up appointments for appropriate PD patients could be considered based on their preferences [35–38]. The development and utilisation of single-point-of-access, integrated models of care, the role of technology in supporting care coordination, increasing speed of communication between services, and the wider use of shared online medical records and electronic prescriptions could break down the barriers of current fragmented care in PD. These suggestions could improve the challenges with appointments and healthcare access experienced in PD. Addressing the workload of medication management reported by PwP and caregivers through regular structured medication reviews, simplification of medication regimes, and ensuring support for PD medication changes with close multidisciplinary team working between PD specialists, GPs and pharmacists was discussed. Furthermore, healthcare professionals could use deprescribing tools such as the STOPP/START criteria and American Geriatrics Society Beers Criteria to reduce medication burden [39]. Clear communication, realistic expectation setting, use of lay terms in clinic letters and normalising experiences of PD symptoms for PwP and caregivers could improve interpersonal relationships between patients and healthcare professionals, and help PwP and caregivers understand and accept the nuances of living with PD.

## Implications for clinical practice

This study highlights the importance for healthcare professionals involved in PD care to understand the concepts of treatment burden and capacity, recognise their potential impact on health outcomes and address these in clinical settings. The MTBQ be a practical tool to help identify PwP and/or caregivers at risk of high treatment burden, enabling timely and proactive interventions. The brevity of the questionnaire makes it feasible for use in clinical settings, although further validation is needed. Attributes such as frailty, PD severity, number of non-motor symptoms and medication frequency should be routinely assessed in PD reviews. Healthcare professionals should consider using patient-centred questions such as those proposed by Mair and May (*"Can you really do what I'm asking you to do?"*) [40], and by Linzer et al (*"What challenges do you experience in your treatment and self-management?"*) to facilitate shared-decision making conversations towards achieving "Minimally Disruptive Medicine", which contain important principles towards the implementation of change at a system level [12,40–42].

Wider utilisation of valuable existing resources including Parkinson's UK, PD support groups, and social prescribing link workers can reduce treatment burden by enabling PwP and caregivers to access tailored information, resources and services. Lower levels of self-management adherence are associated with higher treatment burden levels [43,44]. Furthermore, healthcare professionals have an important role in building trust, increasing motivation and empowering individuals to self-manage their health [45]. A study in patients with end-stage renal failure reported differences in self-management practices, which may be modifiable between those with high and low treatment burden levels [46]. Key self-management components in PD include medication management, completing exercises, symptom monitoring, psychological coping

strategies, maintaining independence and autonomy, engaging with social networks, and obtaining knowledge and information [47]. Capacity coaching, a strategy shown to be feasible within a primary care setting in the USA, could help healthcare professionals co-create strategies with PwP and caregivers to bolster existing sources of capacity or cultivate new capacity for self-management of PD [48,49].

Policy changes to raise awareness and education about treatment burden and capacity in PD amongst healthcare professionals could lead to meaningful change. For example, the 2016 NICE UK multimorbidity guidelines recommended assessing treatment burden alongside an individualised management plan [50]. Yet, a systematic review in 2022 identified only one multi-centre randomised control trial in the UK that measured the effect of primary care service changes across multiple providers on treatment burden using the MTBQ for patients with multimorbidity [51,52]. Our study provides recommendations for interventions in future research studies that could address treatment burden and capacity not just in PD, but also in other long-term conditions.

## Strengths and limitations

The use of the MTBQ to assess the extent and key drivers of treatment burden in PD is an important and novel contribution of this study. However, the MTBQ has not been validated for use amongst caregivers and may be a limitation. This mixed-methods approach enabled the identification of the key drivers of treatment burden through a cross-sectional survey followed by focus groups to develop relevant recommendations for improvement. The survey included a wide sociodemographic range of PwP and caregivers across the UK, with varying lengths of PD diagnosis and severity. Multistakeholder involvement, particularly the inclusion of three PwP and a caregiver in two of the three focus groups alongside healthcare professionals is seen as a strength, allowing discussion from differing perspectives. Regardless, the small number of PwP and caregivers in the focus groups are limitations. The lack of ethnic diversity, the potential for recruitment bias as participants were a self-selected population recruited through Parkinson's UK who had expressed an active interest in participating in research studies and have access to technology, and the small number of caregiver survey respondents may limit the generalisability of findings. Recruitment from PD clinics with the distribution of paper surveys attempted to reduce this bias. Additionally, self-reported data relies heavily on participant recall, and a review of healthcare records should be considered in future studies. Data collection during the COVID-19 pandemic may have had an impact on healthcare service provision and access for PwP and caregivers in the UK. Challenges recruiting PD nurse specialists and general practitioners are a limitation of the focus groups, but perhaps reflect the broader healthcare service constraints in the UK.

## Conclusions

Many PwP and caregivers experienced treatment burden influenced by a multitude of factors. This study has demonstrated that the MTBQ and clinical assessment of frailty, PD severity, non-motor symptoms burden and medication complexity could identify PwP and caregivers at risk of treatment burden. The high treatment burden among caregivers of PwP remains underexplored and warrants further study. Identifying PwP and caregivers who are at risk of treatment burden enables a shift towards the delivery of individualised, holistic patient-centred care in PD. Future research should evaluate the recommended individual- and system-level changes to reduce treatment burden and capacity in PD that could improve overall health outcomes.

## Supporting information

**S1 File. Treatment burden surveys.**
(PDF)

**S2 File. Focus group guide.**
(PDF)

**S3 File. Responses to multimorbidity treatment burden questionnaire.**
(PDF)

## Acknowledgments

The authors would like to thank all participants for taking part in this study and our patient and public involvement group for their invaluable input. We would like to thank Matthew Zimmerman (medical student) for his help in recruiting patients to the survey and assisting with data input. Thank you to Angela Dumbleton, research assistant at the Academic Geriatric Medicine for her administrative support and survey data input. Thank you to Parkinson's UK for their support with participant recruitment. We are also grateful to the developers of the MTBQ for their permission to use the MTBQ and their adapted caregiver version in this study.

## Author contributions

**Conceptualization:** Qian Yue Tan, Kinda Ibrahim, Helen C Roberts, Simon DS Fraser.

**Data curation:** Qian Yue Tan.

**Formal analysis:** Qian Yue Tan.

**Funding acquisition:** Qian Yue Tan, Kinda Ibrahim, Helen C Roberts, Khaled Amar, Simon DS Fraser.

**Investigation:** Qian Yue Tan.

**Methodology:** Qian Yue Tan, Kinda Ibrahim, Helen C Roberts, Khaled Amar, Simon DS Fraser.

**Project administration:** Qian Yue Tan.

**Supervision:** Kinda Ibrahim, Helen C Roberts, Khaled Amar, Simon DS Fraser.

**Validation:** Qian Yue Tan, Kinda Ibrahim, Helen C Roberts, Simon DS Fraser.

**Writing – original draft:** Qian Yue Tan.

**Writing – review & editing:** Qian Yue Tan, Kinda Ibrahim, Khaled Amar, Simon DS Fraser.

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
