## [Decision Letter · Decision Letter 0]

10 Jun 2025

Dear Dr. Tan,

Thank you for submitting your manuscript to PLOS ONE. After careful consideration, we feel that it has merit but does not fully meet PLOS ONE’s publication criteria as it currently stands. Therefore, we invite you to submit a revised version of the manuscript that addresses the points raised during the review process.

We look forward to receiving your revised manuscript.

Kind regards,

Farshid Danesh, Ph.D.

Academic Editor

PLOS ONE

Reviewers' comments:

Reviewer's Responses to Questions

**Comments to the Author**

1. Is the manuscript technically sound, and do the data support the conclusions?

Reviewer #1: Yes

Reviewer #2: Partly

2. Has the statistical analysis been performed appropriately and rigorously?

Reviewer #1: Yes

Reviewer #2: N/A

3. Have the authors made all data underlying the findings in their manuscript fully available?

Reviewer #1: Yes

Reviewer #2: Yes

4. Is the manuscript presented in an intelligible fashion and written in standard English?

Reviewer #1: Yes

Reviewer #2: Yes

Reviewer #1: It is clinically significant to identify factors influencing the burden on PD patients and caregivers, as well as to propose solutions through focused groups.

The following points require responses or actions:

Clarification of the reasoning behind setting the P-value at 0.25

The statistical rationale and significance need to be explained.

Difference in the accuracy of responses between online and paper surveys

Analysis supporting the reliability and consistency of the data should be provided.

Accuracy of PD patients and caregivers in responding to medical information

A detailed explanation of evaluation methods used to assess their understanding and response accuracy is necessary.

MIBQ’s four-category analysis and confirmation of linear relationships with influencing factors

Analysis should elucidate the linear relationships between severity and influencing factors using MIBQ.

Concerns regarding the small number of PD patients in Phase 2 focus groups and generalizability

Addressing the bias and clarifying the representativeness of these patients are essential.

Comparison with similar studies and the novel findings of this study

Clear explanations of the study’s uniqueness, new findings, and their clinical impact are required, along with the perspective of comparison.

Simplification and clarification of the text

The text should be refined to be more concise and clear, improving understanding for readers.

Reviewer #2: This manuscript addresses an important and timely topic — the treatment burden experienced by people with Parkinson’s disease and their caregivers. The mixed-methods approach is generally appropriate, and the study is clearly presented. However, in its current form, the work does not provide sufficient novelty or depth to justify publication.

The survey results largely confirm known associations (e.g., higher burden with greater disease severity, frailty, and polypharmacy). The second phase of the study — the focus groups — is based on a very small and heterogeneous sample (only 11 participants, including just one caregiver), which limits the robustness of the findings. The output of the qualitative work remains purely narrative, without structured outcome measures or any attempt to assess the practical impact of the recommendations.

Moreover, the proposed suggestions, while sensible, are relatively generic and lack strong grounding in the data. Without any follow-up or validation of these recommendations, the contribution remains modest and speculative.

In summary, while the topic is important and the manuscript is well written, the study lacks the scientific relevance and originality required for publication.

**Do you want your identity to be public for this peer review?** For information about this choice, including consent withdrawal, please see our Privacy Policy

Reviewer #1: No

Reviewer #2: No

---

## [Author Response · Author response to Decision Letter 1]

22 Sep 2025

Dear Academic Editor and Reviewers, Thank you for your comments which has helped us improve our manuscript significantly. Please see our response to the comments below and in the attached 'Rebuttal Letter'.

Dear Reviewer #1

- We agree with this comment. We have now included an explanation of the pre-screening stage and justification for the less stringent p-value (p<0.25) for the univariable analysis. This is referenced in the manuscript.

- Thank you for your suggestion. From a total of 192 surveys, there were 31 paper surveys returned. With the small sample of paper surveys, analysis comparing online and paper surveys were not conducted. We have now included this in the manuscript. As stated in Page 5 Lines 113-114, all paper survey responses were manually double entered into the data spreadsheet to ensure accuracy and reliability.

We agree with this comment and recognise this limitation in the manuscript. The use of self-reported data for long-term conditions and variations with medical records was stated in the discussion (Lines 344). We have now expanded this further in the ‘Strengths and Limitation’ section of the paper and included suggestions for future studies.

- Thank you for this comment, the categorical analysis has been determined based on the development of MTBQ questionnaire. . Please see publication: Duncan P, Murphy M, Man M, et al. Development and validation of the Multimorbidity Treatment Burden Questionnaire (MTBQ) BMJ Open 2020;8:e019413. doi: 10.1136/bmjopen-2017-019413. The MTBQ research team recommend the use of four-categories due to skewness of global MTBQ scores, with proportion of patients within each category reported. This has been validated further by the research team in a 2023 publication: Duncan P, Scott LJ, Dawson S, et al.

Further development and validation of the Multimorbidity Treatment Burden Questionnaire (MTBQ)BMJ Open 2024;14:e080096. doi: 10.1136/bmjopen-2023-080096/ As such, analysis depicting the linear relationships between severity of treatment burden and variable was not considered to be appropriate for this study. The justification for the dichotomisation of categories for analysis is described in the Methods.

- Thank you for your comment. Whilst there were three PD patients (out of 12 participants) in the focus groups, the small sample size of PD patients in the focus groups has been added as a limitation in the discussion leading to the lack of generalisability of the study findings. However, focus groups with carers/PD patients alongside healthcare professionals should be seen as a strength given the opportunity to discuss varying views and experiences.

- Thank you, we have further emphasised the novel findings and comparisons with previous literature in the discussion section. We have also included a subsection on ‘Implications for Clinical Practice’ in the Discussion.

- Thank you, we have amended sentences to ensure it is concise and clear where appropriate.

Dear Reviewer #2,

- Thank you, we agree that this is an important and timely topic. Treatment burden specific to Parkinson’s and associated factors have not be quantified prior to this study and is therefore a novel finding. Equally the association with frailty and treatment burden has not been identified in previous studies.

Our study highlights that treatment burden in commonly experienced and could therefore lead to poor outcomes in this population. 84% of people with Parkinson’s and 80% of caregivers reported treatment burden. This study highlights key determines of treatment burden, with recommendations for change in clinical practice that can reduce this. We have amended the discussion section to describe specific recommendations for change in clinical practice.

Following this, we believe the updated paper addressing the previous reviewers’ comments has strengthened the paper, highlighting its novelty with clear merits for publication.

- Thank you for this comment. We have amended the discussion section to improve clarity on how the associations found in this study that are specific to treatment burden levels in PD. We have included a subsection that includes implications of our findings that can lead to practical change in clinical practice.

We recognise the limitations the reviewer mentions but, as per our responses to other reviewer comments we have further highlighted the areas of novelty in this under-researched field. Furthermore, treatment burden in patients with long-term conditions remains an under-researched area and this adds valuably to the literature.

- Thank you for your comment. We have made significant changes to the discussion section in response to the reviewer comments above that address these issues.

---

## [Editor Report · Decision Letter 1]

2 Oct 2025

Dear Dr. Tan,

We look forward to receiving your revised manuscript.

Kind regards,

Farshid Danesh, Ph.D.

Academic Editor

PLOS ONE

Journal Requirements:

Additional Editor Comments:

The respected authors have incorporated the reviewers’ edits and comments to the fullest extent possible and have revised the article accordingly. However, it is imperative to emphasize that the manuscript still requires comprehensive linguistic, grammatical, and stylistic editing. The authors are strongly encouraged to undertake a thorough and professional revision to ensure the text adheres to the highest standards of academic writing and clarity.

---

## [Author Response · Author response to Decision Letter 2]

21 Nov 2025

Confirmation of Journal requirements:

We have reviewed the reviewers comments once again and confirm that there have been no recommendations to cite specific previously published works. The additional references (26,26) was added in response to comment No.4 below to justify the rationale of p-value as requested.

We have reviewed the reference list and can confirm that no cited papers have been retracted.

We have removed the following reference on review as it was not required in the discussion section: Eton DT, Ridgeway JL, Linzer M, Boehm DH, Rogers EA, Yost KJ, et al. Healthcare provider relational quality is associated with better self-management and less treatment burden in people with multiple chronic conditions. Patient Prefer Adherence. 2017;11:1635–46. Epub 2017/10/17. doi: 10.2147/ppa.S145942. PubMed PMID: 29033551; PubMed Central PMCID: PMCPMC5630069.

Response to Additional Editor Comments:

Thank you, we have worked hard to incorporate the reviewers’ comments and believe that the manuscript has now been strengthened further.

---

## [Editor Report · Decision Letter 2]

26 Nov 2025

A mixed-methods study to explore the modifiable aspects of treatment burden in Parkinson’s disease and develop recommendations for improvement.

PONE-D-24-41305R2

Dear Dr. Tan,

We’re pleased to inform you that your manuscript has been judged scientifically suitable for publication and will be formally accepted for publication once it meets all outstanding technical requirements.

Kind regards,

Farshid Danesh, Ph.D.

Academic Editor

PLOS ONE
---

## [Editor Report · Acceptance letter]

PONE-D-24-41305R2

PLOS One

Dear Dr. Tan,

I'm pleased to inform you that your manuscript has been deemed suitable for publication in PLOS One. Congratulations! Your manuscript is now being handed over to our production team.

Kind regards,

on behalf of

Associate Professor Farshid Danesh

Academic Editor

PLOS One